# Learning with Average Top-k Loss

**Yanbo Fan**[3,4,1] , **Siwei Lyu**[1*], **Yiming Ying**[2] , **Bao-Gang Hu**[3,4]
[1]Department of Computer Science, University at Albany, SUNY
[2]Department of Mathematics and Statistics, University at Albany, SUNY
[3]National Laboratory of Pattern Recognition, CASIA
[4]University of Chinese Academy of Sciences (UCAS)
{yanbo.fan,hubg}@nlpr.ia.ac.cn, slyu@albany.edu, yying@albany.edu

## Abstract

In this work, we introduce the *average top-$k$* ($\mathrm{AT}_k$) loss as a new aggregate loss for supervised learning, which is the average over the $k$ largest individual losses over a training dataset. We show that the $\mathrm{AT}_k$ loss is a natural generalization of the two widely used aggregate losses, namely the average loss and the maximum loss, but can combine their advantages and mitigate their drawbacks to better adapt to different data distributions. Furthermore, it remains a convex function over all individual losses, which can lead to convex optimization problems that can be solved effectively with conventional gradient-based methods. We provide an intuitive interpretation of the $\mathrm{AT}_k$ loss based on its equivalent effect on the continuous individual loss functions, suggesting that it can reduce the penalty on correctly classified data. We further give a learning theory analysis of $\mathrm{MAT}_k$ learning on the classification calibration of the $\mathrm{AT}_k$ loss and the error bounds of $\mathrm{AT}_k$-SVM. We demonstrate the applicability of minimum average top-$k$ learning for binary classification and regression using synthetic and real datasets.

## 1 Introduction

Supervised learning concerns the inference of a function $f : \mathcal{X} \mapsto \mathcal{Y}$ that predicts a target $y \in \mathcal{Y}$ from data/features $\mathbf{x} \in \mathcal{X}$ using a set of labeled training examples $\{(\mathbf{x}_i, y_i)\}_{i=1}^n$. This is typically achieved by seeking a function $f$ that minimizes an *aggregate loss* formed from *individual losses* evaluated over all training samples.

To be more specific, the individual loss for a sample $(\mathbf{x}, y)$ is given by $\ell(f(\mathbf{x}), y)$, in which $\ell$ is a nonnegative bivariate function that evaluates the quality of the prediction made by function $f$. For example, for binary classification (*i.e.*, $y_i \in \{\pm 1\}$), commonly used forms for individual loss include the 0-1 loss, $\mathbb{I}_{yf(\mathbf{x}) \leq 0}$, which is 1 when $y$ and $f(\mathbf{x})$ have different sign and 0 otherwise, the hinge loss, $\max(0, 1 - yf(\mathbf{x}))$, and the logistic loss, $\log_2(1 + \exp(-yf(\mathbf{x})))$, all of which can be further simplified as the so-called *margin loss*, *i.e.*, $\ell(y, f(\mathbf{x})) = \ell(yf(\mathbf{x}))$. For regression, squared difference $(y - f(\mathbf{x}))^2$ and absolute difference $|y - f(\mathbf{x})|$ are two most popular forms for individual loss, which can be simplified as $\ell(y, f(\mathbf{x})) = \ell(|y - f(\mathbf{x})|)$. Usually the individual loss is chosen to be a convex function of its input, but recent works also propose various types of non-convex individual losses (*e.g.*, [10, 15, 27, 28]).

The supervised learning problem is then formulated as $\min_f \{\mathcal{L}(L_{\mathbf{z}}(f)) + \Omega(f)\}$, where $\mathcal{L}(L_{\mathbf{z}}(f))$ is the aggregate loss accumulates all individual losses over training samples, *i.e.*, $L_{\mathbf{z}}(f) = \{\ell_i(f)\}_{i=1}^n$, with $\ell_i(f)$ being the shorthand notation for $\ell(f(\mathbf{x}_i), y_i)$, and $\Omega(f)$ is the regularizer on $f$. However, in contrast to the plethora of the types of individual losses, there are only a few choices when we consider the aggregate loss:

---

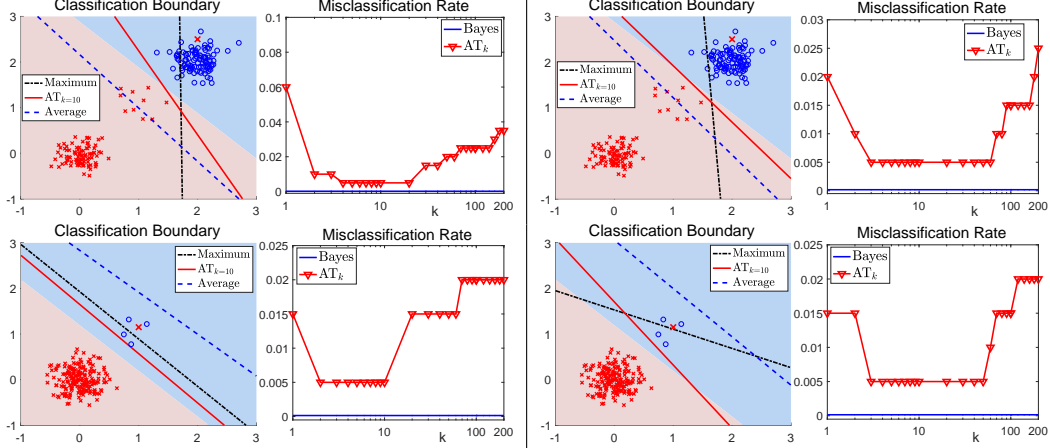

Figure 1: *Comparison of different aggregate losses on 2D synthetic datasets with $n = 200$ samples for binary classification on a balanced but multi-modal dataset and with outliers (**top**) and an imbalanced dataset with outliers (**bottom**) with logistic loss (**left**) and hinge loss (**right**). Outliers in data are shown as an enlarged $\times$ and the optimal Bayes classifications are shown as shaded areas. The figures in the second and fourth columns show the misclassification rate of $AT_k$ vs. $k$ for each case.*

- the *average loss*: $\mathcal{L}_{\text{avg}}(L_{\mathbf{z}}(f)) = \frac{1}{n}\sum_{i=1}^{n}\ell_i(f)$, *i.e.*, the mean of all individual losses;
- the *maximum loss*: $\mathcal{L}_{\max}(L_{\mathbf{z}}(f)) = \max_{1 \le k \le n}\ell_i(f)$, *i.e.*, the largest individual loss;
- the *top-k loss* [20]: $\mathcal{L}_{\text{top-}k}(L_{\mathbf{z}}(f)) = \ell_{[k]}(f)^2$ for $1 \le k \le n$, *i.e.*, the $k$-th largest (top-$k$) individual loss.

The average loss is unarguably the most widely used aggregate loss, as it is a unbiased approximation to the expected risk and leads to the *empirical risk minimization* in learning theory [1, 7, 22, 25, 26]. Further, minimizing the average loss affords simple and efficient stochastic gradient descent algorithms [3, 21]. On the other hand, the work in [20] shows that constructing learning objective based on the maximum loss may lead to improved performance for data with separate typical and rare subpopulations. The top-k loss [20] generalizes the maximum loss, as $\mathcal{L}_{\max}(L_{\mathbf{z}}(f)) = \mathcal{L}_{\text{top-1}}(L_{\mathbf{z}}(f))$, and can alleviate the sensitivity to outliers of the latter. However, unlike the average loss or the maximum loss, the top-k loss in general does not lead to a convex learning objective, as it is not convex of all the individual losses $L_{\mathbf{z}}(f)$.

In this work, we propose a new type of aggregate loss that we term as the *average top-k* ($AT_k$) loss, which is the average of the largest $k$ individual losses, that is defined as:

$$\mathcal{L}_{\text{avt-}k}(L_{\mathbf{z}}(f)) = \frac{1}{k}\sum_{i=1}^{k}\ell_{[i]}(f). \tag{1}$$

We refer to learning objectives based on minimizing the $AT_k$ loss as *$MAT_k$ learning*.

The $AT_k$ loss generalizes the average loss ($k = n$) and the maximum loss ($k = 1$), yet it is less susceptible to their corresponding drawbacks, *i.e.*, it is less sensitive to outliers than the maximum loss and can adapt to imbalanced and/or multi-modal data distributions better than the average loss. This is illustrated with two toy examples of synthesized 2D data for binary classification in Fig.1 (see *supplementary materials* for a complete illustration). As these plots show, the linear classifier obtained with the maximum loss is not optimal due to the existence of outliers while the linear classifier corresponding to the average loss has to accommodate the requirement to minimize individual losses across all training data, and sacrifices smaller sub-clusters of data (*e.g.*, the rare population of $+$ class in the top row and the smaller dataset of $-$ class in the bottom row). In contrast, using $AT_k$ loss with $k = 10$ can better protect such smaller sub-clusters and leads to linear classifiers closer to the optimal Bayesian linear classifier. This is also corroborated by the plots of corresponding misclassification rate of $AT_k$ vs. $k$ value in Fig.1, which show that minimum misclassification rates occur at $k$ value other than 1 (maximum loss) or $n$ (average loss).

The $AT_k$ loss is a tight upper-bound of the top-k loss, as $\mathcal{L}_{\text{avt-}k}(L_{\mathbf{z}}(f)) \ge \mathcal{L}_{\text{top-}k}(L_{\mathbf{z}}(f))$ with equality holds when $k = 1$ or $\ell_i(f) = $ constant, and it is a convex function of the individual losses (see Section 2). Indeed, we can express $\ell_{[k]}(f)$ as the difference of two convex functions $k\mathcal{L}_{\text{avt-}k}(L_{\mathbf{z}}(f)) - (k-1)\mathcal{L}_{\text{avt-}(k-1)}(L_{\mathbf{z}}(f))$, which shows that in general $\mathcal{L}_{\text{top-}k}(L_{\mathbf{z}}(f))$ is not convex with regards to the individual losses.

In sequel, we will provide a detailed analysis of the $AT_k$ loss and $MAT_k$ learning. First, we establish a reformulation of the $AT_k$ loss as the minimum of the average of the individual losses over all training examples transformed by a hinge function. This reformulation leads to a simple and effective stochastic gradient-based algorithm for $MAT_k$ learning, and interprets the effect of the $AT_k$ loss as shifting down and truncating at zero the individual loss to reduce the undesirable penalty on correctly classified data. When combined with the hinge function as individual loss, the $AT_k$ aggregate loss leads to a new variant of SVM algorithm that we term as $AT_k$ SVM, which generalizes the C-SVM and the $\nu$-SVM algorithms [19]. We further study learning theory of $MAT_k$ learning, focusing on the classification calibration of the $AT_k$ loss function and error bounds of the $AT_k$ SVM algorithm. This provides a theoretical lower-bound for $k$ for reliable classification performance. We demonstrate the applicability of minimum average top-$k$ learning for binary classification and regression using synthetic and real datasets.

The main contributions of this work can be summarized as follows.

- We introduce the $AT_k$ loss for supervised learning, which can balance the pros and cons of the average and maximum losses, and allows the learning algorithm to better adapt to imbalanced and multi-modal data distributions.
- We provide algorithm and interpretation of the $AT_k$ loss, suggesting that most existing learning algorithms can take advantage of it without significant increase in computation.
- We further study the theoretical aspects of $AT_k$ loss on classification calibration and error bounds of minimum average top-$k$ learning for $AT_k$-SVM.
- We perform extensive experiments to validate the effectiveness of the $MAT_k$ learning.

## 2   Formulation and Interpretation

The original $AT_k$ loss, though intuitive, is not convenient to work with because of the sorting procedure involved. This also obscures its connection with the statistical view of supervised learning as minimizing the expectation of individual loss with regards to the underlying data distribution. Yet, it affords an equivalent form, which is based on the following result.

**Lemma 1** (Lemma 1, [16]). $\sum_{i=1}^{k} x_{[i]}$ *is a convex function of* $(x_1, \cdots, x_n)$. *Furthermore, for* $x_i \geq 0$ *and* $i = 1, \cdots, n$, *we have* $\sum_{i=1}^{k} x_{[i]} = \min_{\lambda \geq 0} \left\{ k\lambda + \sum_{i=1}^{n} [x_i - \lambda]_+ \right\}$, *where* $[a]_+ = \max\{0, a\}$ *is the hinge function.*

For completeness, we include a proof of Lemma 1 in *supplementary materials*. Using Lemma 1, we can reformulate the $AT_k$ loss (1) as

$$\mathcal{L}_{\text{avt-}k}(L_{\mathbf{z}}(f)) = \frac{1}{k} \sum_{i=1}^{k} \ell_{[i]}(f) \propto \min_{\lambda \geq 0} \left\{ \frac{1}{n} \sum_{i=1}^{n} [\ell_i(f) - \lambda]_+ + \frac{k}{n} \lambda \right\}. \tag{2}$$

In other words, the $AT_k$ loss is equivalent to minimum of the average of individual losses that are shifted and truncated by the hinge function controlled by $\lambda$. This sheds more lights on the $AT_k$ loss, which is particularly easy to illustrate in the context of binary classification using the margin losses, $\ell(f(\mathbf{x}), y) = \ell(yf(\mathbf{x}))$.

In binary classification, the "gold standard" of individual loss is the 0-1 loss $\mathbb{I}_{yf(\mathbf{x}) \leq 0}$, which exerts a constant penalty 1 to examples that are misclassified by $f$ and no penalty to correctly classified examples. However, the 0-1 loss is difficult to work as it is neither continuous nor convex. In practice, it is usually replaced by a surrogate convex loss. Such convex surrogates afford efficient algorithms, but as continuous and convex upper-bounds of the 0-1 loss, they typically also penalize correctly classified examples, *i.e.*, for $y$ and $\mathbf{x}$ that satisfy $yf(\mathbf{x}) > 0$, $\ell(yf(\mathbf{x})) > 0$, whereas $\mathbb{I}_{yf(\mathbf{x}) \leq 0} = 0$ (Fig.2). This implies that when the average of individual losses across all training examples is minimized, correctly classified examples by $f$ that are "too close" to the classification boundary may be sacrificed to accommodate reducing the average loss, as is shown in Fig.1.

In contrast, after the individual loss is combined with the hinge function, *i.e.*, $[\ell(yf(\mathbf{x})) - \lambda]_+$ with $\lambda > 0$, it has the effect of "shifting down" the original individual loss function and truncating it at zero, see Fig.2. The transformation of the individual loss reduces penalties of all examples, and in particular benefits correctly classified data. In particular, if such examples are "far enough" from the decision boundary, like in the 0-1 loss, their penalty becomes zero. This alleviates the likelihood of misclassification on those rare sub-populations of data that are close to the decision boundary.

**Algorithm**: The reformulation of the $AT_k$ loss in Eq.(2) also facilitates development of optimization algorithms for the minimum $AT_k$ learning. As practical supervised learning problems usually use a parametric form of $f$, as $f(\mathbf{x}; \mathbf{w})$, where $\mathbf{w}$ is the parameter, the corresponding minimum $AT_k$ objective becomes

$$\min_{\mathbf{w}, \lambda \geq 0} \left\{ \frac{1}{n} \sum_{i=1}^{n} [\ell(f(\mathbf{x}_i; \mathbf{w}), y_i) - \lambda]_+ + \frac{k}{n}\lambda + \Omega(\mathbf{w}) \right\},$$
(3)

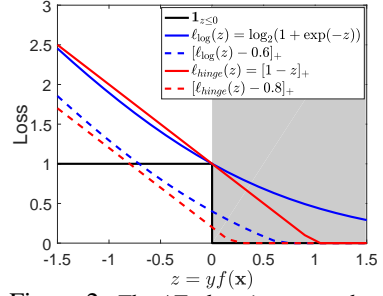

Figure 2: *The $AT_k$ loss interpreted at the individual loss level. Shaded area corresponds to data/target with correct classification.*

It is not hard to see that if $\ell(f(\mathbf{x}; \mathbf{w}), y)$ is convex with respect to $\mathbf{w}$, the objective function of in Eq.(3) is a convex function for $\mathbf{w}$ and $\lambda$ jointly. This leads to an immediate stochastic (projected) gradient descent [3, 21] for solving (3). For instance, with $\Omega(\mathbf{w}) = \frac{1}{2C}\|\mathbf{w}\|^2$, where $C > 0$ is a regularization factor, at the $t$-th iteration, the corresponding $MAT_k$ objective can be minimized by first randomly sampling $(\mathbf{x}_{i_t}, y_{i_t})$ from the training set and then updating the parameters as

$$
\begin{aligned}
\mathbf{w}^{(t+1)} &\leftarrow \mathbf{w}^{(t)} - \eta_t \left( \partial_{\mathbf{w}} \ell(f(\mathbf{x}_{i_t}; \mathbf{w}^{(t)}), y_{i_t}) \cdot \mathbb{I}_{[\ell(f(\mathbf{x}_{i_t}; \mathbf{w}^{(t)}), y_{i_t}) > \lambda^{(t)}]} + \frac{\mathbf{w}^{(t)}}{C} \right) \\
\lambda^{(t+1)} &\leftarrow \left[ \lambda^{(t)} - \eta_t \left( \frac{k}{n} - \mathbb{I}_{[\ell(f(\mathbf{x}_{i_t}; \mathbf{w}^{(t)}, y_{i_t}) > \lambda^{(t)}]} \right) \right]_+
\end{aligned}
$$
(4)

where $\partial_{\mathbf{w}} \ell(f(\mathbf{x}; \mathbf{w}), y)$ denotes the sub-gradient with respect to $\mathbf{w}$, and $\eta_t \sim \frac{1}{\sqrt{t}}$ is the step size.

**$AT_k$-SVM**: As a general aggregate loss, the $AT_k$ loss can be combined with any functional form for individual losses. In the case of binary classification, the $AT_k$ loss combined with the individual hinge loss for a prediction function $f$ from a reproducing kernel Hilbert space (RKHS) [18] leads to the $AT_k$-SVM model. Specifically, we consider function $f$ as a member of RKHS $\mathcal{H}_K$ with norm $\|\cdot\|_K$, which is induced from a reproducing kernel $K : \mathcal{X} \times \mathcal{X} \to \mathbb{R}$. Using the individual hinge loss, $[1 - y_i f(\mathbf{x}_i)]_+$, the corresponding $MAT_k$ learning objective in RKHS becomes

$$\min_{f \in \mathcal{H}_K, \lambda \geq 0} \frac{1}{n} \sum_{i=1}^{n} \left[ [1 - y_i f(\mathbf{x}_i)]_+ - \lambda \right]_+ + \frac{k}{n}\lambda + \frac{1}{2C}\|f\|_K^2,$$
(5)

where $C > 0$ is the regularization factor. Furthermore, the outer hinge function in (5) can be removed due to the following result.

**Lemma 2.** *For $a \geq 0$, $b \geq 0$, there holds $\left[ [a - \ell]_+ - b \right]_+ = [a - b - \ell]_+$.*

Proof of Lemma 2 can be found in the *supplementary materials*. In addition, note that for any minimizer $(f_\mathbf{z}, \lambda_\mathbf{z})$ of (5), setting $f(\mathbf{x}) = 0$, $\lambda = 1$ in the objective function of (5), we have $\frac{k}{n}\lambda_\mathbf{z} \leq \frac{1}{n}\sum_{i=1}^{n} \left[ [1 - y_i f_\mathbf{z}(\mathbf{x}_i)]_+ - \lambda_\mathbf{z} \right]_+ + \frac{k}{n}\lambda_\mathbf{z} + \frac{1}{2C}\|f_\mathbf{z}\|_K^2 \leq \frac{k}{n}$, so we have $0 \leq \lambda_\mathbf{z} \leq 1$ which means that the minimization can be restricted to $0 \leq \lambda \leq 1$. Using these results and introducing $\rho = 1 - \lambda$, Eq.(5) can be rewritten as

$$\min_{f \in \mathcal{H}_K, 0 \leq \rho \leq 1} \frac{1}{n} \sum_{i=1}^{n} [\rho - y_i f(\mathbf{x}_i)]_+ - \frac{k}{n}\rho + \frac{1}{2C}\|f\|_K^2.$$
(6)

The $AT_k$-SVM objective generalizes many several existing SVM models. For example, when $k = n$, it equals to the standard C-SVM [5]. When $C = 1$ and with conditions $K(\mathbf{x}_i, \mathbf{x}_i) \leq 1$ for any $i$, $AT_k$-SVM reduces to $\nu$-SVM [19] with $\nu = \frac{k}{n}$. Furthermore, similar to the conventional SVM model, writing in the dual form of (6) can lead to a convex quadratic programming problem that can be solved efficiently. See *supplementary materials* for more detailed explanations.

**Choosing $k$.** The number of top individual losses in the $AT_k$ loss is a critical parameter that affects the learning performance. In concept, using $AT_k$ loss will not be worse than using average or maximum losses as they correspond to specific choices of $k$. In practice, $k$ can be chosen during training from a validation dataset as the experiments in Section 4. As $k$ is an integer, a simple grid search usually suffices to find a satisfactory value. Besides, Theorem 1 in Section 3 establishes a theoretical lower bound for $k$ to guarantee reliable classification based on the Bayes error. If we have information about the proportion of outliers, we can also narrow searching space of $k$ based on the fact that $AT_k$ loss is the convex upper bound of the top-k loss, which is similar to [20].

# 3 Statistical Analysis

In this section, we address the statistical properties of the $\text{AT}_k$ objective in the context of binary classification. Specifically, we investigate the property of *classification calibration* [1] of the $\text{AT}_k$ general objective, and derive bounds for the misclassification error of the $\text{AT}_k$-SVM model in the framework of statistical learning theory (*e.g.* [1, 7, 23, 26]).

## 3.1 Classification Calibration under $\text{AT}_k$ Loss

We assume the training data $\mathbf{z} = \{(\mathbf{x}_i, y_i)\}_{i=1}^n$ are i.i.d. samples from an unknown distribution $p$ on $\mathcal{X} \times \{\pm 1\}$. Let $p_{\mathcal{X}}$ be the marginal distribution of $p$ on the input space $\mathcal{X}$. Then, the misclassification error of a classifier $f : \mathcal{X} \to \{\pm 1\}$ is denoted by $\mathcal{R}(f) = \Pr(y \neq f(\mathbf{x})) = \mathbb{E}[\mathbb{I}_{yf(\mathbf{x}) \leq 0}]$. The Bayes error is given by $\mathcal{R}^* = \inf_f \mathcal{R}(f)$, where the infimum is over all measurable functions. No function can achieve less risk than the Bayes rule $f_c(\mathbf{x}) = \text{sign}(\eta(\mathbf{x}) - \frac{1}{2})$, where $\eta(\mathbf{x}) = \Pr(y = 1|\mathbf{x})$ [8].

In practice, one uses a surrogate loss $\ell : \mathbb{R} \to [0, \infty)$ which is convex and upper-bound the 0-1 loss. The population $\ell$-risk (generalization error) is given by $\mathcal{E}_\ell(f) = \mathbb{E}[\ell(yf(x))]$. Denote the optimal $\ell$-risk by $\mathcal{E}_\ell^* = \inf_f \mathcal{E}_\ell(f)$. A very basic requirement for using such a surrogate loss $\ell$ is the so-called *classification calibration* (point-wise form of Fisher consistency) [1, 14]. Specifically, a loss $\ell$ is *classification calibrated* with respect to distribution $p$ if, for any $x$, the minimizer $f_\ell^* = \inf_f \mathcal{E}_\ell(f)$ should have the same sign as the Bayes rule $f_c(\mathbf{x})$, *i.e.*, $\text{sign}(f_\ell^*(\mathbf{x})) = \text{sign}(f_c(\mathbf{x}))$ whenever $f_c(\mathbf{x}) \neq 0$.

An appealing result concerning the classification calibration of a loss function $\ell$ was obtained in [1], which states that $\ell$ is classification calibrated if $\ell$ is convex, differentiable at 0 and $\ell'(0) < 0$. In the same spirit, we investigate the classification calibration property of the $\text{AT}_k$ loss. Specifically, we first obtain the population form of the $\text{AT}_k$ objective using the infinite limit of (2)

$$\frac{1}{n} \sum_{i=1}^n [\ell(y_i f(\mathbf{x}_i)) - \lambda]_+ + \frac{k}{n} \lambda \xrightarrow[n \to \infty]{\frac{k}{n} \to \nu} \mathbb{E}[[\ell(yf(\mathbf{x})) - \lambda]_+] + \nu\lambda.$$

We then consider the optimization problem

$$(f^*, \lambda^*) = \arg \inf_{f, \lambda \geq 0} \mathbb{E}[[\ell(yf(\mathbf{x})) - \lambda]_+] + \nu\lambda, \tag{7}$$

where the infimum is taken over all measurable function $f : \mathcal{X} \to \mathbb{R}$. We say the $\text{AT}_k$ (aggregate) loss is classification calibrated with respect to $p$ if $f^*$ has the same sign as the Bayes rule $f_c$. The following theorem establishes such conditions.

**Theorem 1.** *Suppose the individual loss $\ell : \mathbb{R} \to \mathbb{R}^+$ is convex, differentiable at 0 and $\ell'(0) < 0$. Without loss of generality, assume that $\ell(0) = 1$. Let $(f^*, \lambda^*)$ be defined in (7),*

    *(i) If $\nu > \mathcal{E}_\ell^*$ then the $\text{AT}_k$ loss is classification calibrated.*

    *(ii) If, moreover, $\ell$ is monotonically decreasing and the $\text{AT}_k$ aggregate loss is classification calibrated then $\nu \geq \int_{\eta(\mathbf{x}) \neq \frac{1}{2}} \min(\eta(\mathbf{x}), 1 - \eta(\mathbf{x})) dp_{\mathcal{X}}(\mathbf{x})$.*

The proof of Theorem 1 can be found in the *supplementary materials*. Part (i) and (ii) of the above theorem address respectively the sufficient and necessary conditions on $\nu$ such that the $\text{AT}_k$ loss becomes classification calibrated. Since $\ell$ is an upper bound surrogate of the 0-1 loss, the optimal $\ell$-risk $\mathcal{E}_\ell^*$ is larger than the Bayes error $\mathcal{R}^*$, *i.e.*, $\mathcal{E}_\ell^* \geq \mathcal{R}^*$. In particular, if the individual loss $\ell$ is the hinge loss then $\mathcal{E}_\ell^* = 2\mathcal{R}^*$. Part (ii) of the above theorem indicates that the $\text{AT}_k$ aggregate loss is classification calibrated if $\nu = \lim_{n \to \infty} k/n$ is larger than the optimal generalization error $\mathcal{E}_\ell^*$ associated with the individual loss. The choice of $k > n\mathcal{E}_\ell^*$ thus guarantees classification calibration, which gives a lower bound of $k$. This result also provides a theoretical underpinning of the sensitivity to outliers of the maximum loss ($\text{AT}_k$ loss with $k = 1$). If the probability of the set $\{x : \eta(x) = 1/2\}$ is zero, $\mathcal{R}^* = \int_{\mathcal{X}} \min(\eta(\mathbf{x}), 1 - \eta(\mathbf{x})) dp_{\mathcal{X}}(\mathbf{x}) = \int_{\eta(\mathbf{x}) \neq 1/2} \min(\eta(\mathbf{x}), 1 - \eta(\mathbf{x})) dp_{\mathcal{X}}(x)$. Theorem 1 indicates that in this case, if the maximum loss is calibrated, one must have $\frac{1}{n} \approx \nu \geq R^*$. In other words, as the number of training data increases, the Bayes error has to be arbitrarily small, which is consistent with the empirical observation that the maximum loss works well under the well-separable data setting but are sensitive to outliers and non-separable data.

## 3.2 Error bounds of $AT_k$-SVM

We next study the excess misclassification error of the $AT_k$-SVM model *i.e.*, $\mathcal{R}(\text{sign}(f_\mathbf{z})) - \mathcal{R}^*$. Let $(f_\mathbf{z}, \rho_\mathbf{z})$ be the minimizer of the $AT_k$-SVM objective (6) in the RKHS setting. Let $f_\mathcal{H}$ be the minimizer of the generalization error over the RKHS space $\mathcal{H}_K$, *i.e.*, $f_\mathcal{H} = \text{argmin}_{f \in \mathcal{H}_K} \mathcal{E}_h(f)$, where we use the notation $\mathcal{E}_h(f) = \mathbb{E}\left[[1 - yf(\mathbf{x})]_+\right]$ to denote the $\ell$-risk of the hinge loss. In the finite-dimension case, the existence of $f_\mathcal{H}$ follows from the direct method in the variational calculus, as $\mathcal{E}_h(\cdot)$ is lower bounded by zero, coercive, and weakly sequentially lower semi-continuous by its convexity. For an infinite dimensional $\mathcal{H}_K$, we assume the existence of $f_\mathcal{H}$. We also assume that $\mathcal{E}_h(f_\mathcal{H}) < 1$ since even a naïve zero classifier can achieve $\mathcal{E}_h(0) = 1$. Denote the approximation error by $\mathcal{A}(\mathcal{H}_K) = \inf_{f \in \mathcal{H}_K} \mathcal{E}_h(f) - \mathcal{E}_h(f_c) = \mathcal{E}_h(f_\mathcal{H}) - \mathcal{E}_h(f_c)$, and let $\kappa = \sup_{\mathbf{x} \in \mathcal{X}} \sqrt{K(\mathbf{x}, \mathbf{x})}$. The main theorem can be stated as follows.

**Theorem 2.** *Consider the $AT_k$-SVM in RKHS (6). For any $\varepsilon \in (0, 1]$ and $\mu \in (0, 1 - \mathcal{E}_h(f_\mathcal{H}))$, choosing $k = \lceil n(\mathcal{E}_h(f_\mathcal{H}) + \mu) \rceil$. Then, it holds*

$$\Pr\left\{\mathcal{R}(\text{sign}(f_\mathbf{z})) - \mathcal{R}^* \geq \mu + \mathcal{A}(\mathcal{H}) + \varepsilon + \frac{1 + C_{\kappa, \mathcal{H}}}{\sqrt{n}\mu}\right\} \leq 2\exp\left(-\frac{n\mu^2\varepsilon^2}{(1 + C_{\kappa, \mathcal{H}})^2}\right),$$

*where $C_{\kappa, \mathcal{H}} = \kappa(2\sqrt{2C} + 4\|f_\mathcal{H}\|_K)$.*

The complete proof of Theorem 2 is given in the *supplementary materials*. The main idea is to show that $\rho_\mathbf{z}$ is bounded from below by a positive constant with high probability, and then bound the excess misclassification error $\mathcal{R}(\text{sign}(f_\mathbf{z}^*)) - \mathcal{R}^*$ by $\mathcal{E}_h(f_\mathbf{z}/\rho_\mathbf{z}) - \mathcal{E}_h(f_c)$. If $K$ is a universal kernel then $\mathcal{A}(\mathcal{H}_K) = 0$ [23]. In this case, let $\mu = \varepsilon \in (0, 1 - \mathcal{E}_h(f_\mathcal{H}))$, then from Theorem 2 we have

$$\Pr\left\{\mathcal{R}(\text{sign}(f_\mathbf{z})) - \mathcal{R}^* \geq 2\varepsilon + \frac{1 + C_{\kappa, \mathcal{H}}}{\sqrt{n}\varepsilon}\right\} \leq 2\exp\left(-\frac{n\varepsilon^4}{(1 + C_{\kappa, \mathcal{H}})^2}\right),$$

Consequently, choosing $C$ such that $\lim_{n \to \infty} C/n = 0$, which is equivalent to $\lim_{n \to \infty} (1 + C_{\kappa, \mathcal{H}})^2/n = 0$, then $\mathcal{R}(\text{sign}(f_\mathbf{z}))$ can be arbitrarily close to the Bayes error $\mathcal{R}^*$, with high probability, as long as $n$ is sufficiently large.

## 4 Experiments

We have demonstrated that $AT_k$ loss provides a continuum between the average loss and the maximum loss, which can potentially alleviates their drawbacks. A natural question is whether such an advantage actually benefits practical learning problems. In this section, we demonstrate the behaviors of $MAT_k$ learning coupled with different individual losses for binary classification and regression on synthetic and real datasets, with minimizing the average loss and the maximum loss treated as special cases for $k = n$ and $k = 1$, respectively. For simplicity, in all experiments, we use homogenized linear prediction functions $f(\mathbf{x}) = \mathbf{w}^T \mathbf{x}$ with parameters $\mathbf{w}$ and the Tikhonov regularizer $\Omega(\mathbf{w}) = \frac{1}{2C}\|\mathbf{w}\|^2$, and optimize the $MAT_k$ learning objective with the stochastic gradient descent method given in (4).

**Binary Classification:** We conduct experiments on binary classification using eight benchmark datasets from the UCI[3] and KEEL[4] data repositories to illustrate the potential effects of using $AT_k$ loss in practical learning to adapt to different underlying data distributions. A detailed description of the datasets is given in *supplementary materials*. The standard individual logistic loss and hinge loss are combined with different aggregate losses. Note that average loss combined with individual logistic loss corresponds to the logistic regression model and average loss combined with individual hinge loss leads to the C-SVM algorithm [5].

For each dataset, we randomly sample 50%, 25%, 25% examples as training, validation and testing sets, respectively. During training, we select parameters $C$ (regularization factor) and $k$ (number of top losses) on the validation set. Parameter $C$ is searched on grids of $\log_{10}$ scale in the range of $[10^{-5}, 10^5]$ (extended when optimal value is on the boundary), and $k$ is searched on grids of $\log_{10}$ scale in the range of $[1, n]$. We use $k^*$ to denote the optimal $k$ selected from the validation set.

| | Logistic Loss | | | Hinge Loss | | |
|---|---|---|---|---|---|---|
| | Maximum | Average | $AT_{k^*}$ | Maximum | Average | $AT_{k^*}$ |
| Monk | 22.41(2.95) | 20.46(2.02) | **16.76(2.29)** | 22.04(3.08) | 18.61(3.16) | 17.04(2.77) |
| Australian | 19.88(6.64) | 14.27(3.22) | **11.70(2.82)** | 19.82(6.56) | 14.74(3.10) | 12.51(4.03) |
| Madelon | 47.85(2.51) | 40.68(1.43) | **39.65(1.72)** | 48.55(1.97) | 40.58(1.86) | 40.18(1.64) |
| Splice | 23.57(1.93) | 17.25(0.93) | **16.12(0.97)** | 23.40(2.10) | 16.25(1.12) | 16.23(0.97) |
| Spambase | 21.30(3.05) | 8.36(0.97) | 8.36(0.97) | 21.03(3.26) | **7.40(0.72)** | **7.40(0.72)** |
| German | 28.24(1.69) | 25.36(1.27) | **23.28(1.16)** | 27.88(1.61) | 24.16(0.89) | 23.80(1.05) |
| Titanic | 26.50(3.35) | 22.77(0.82) | 22.44(0.84) | 25.45(2.52) | 22.82(0.74) | **22.02(0.77)** |
| Phoneme | 28.67(0.58) | 25.50(0.88) | 24.17(0.89) | 28.81(0.62) | **22.88(1.01)** | **22.88(1.01)** |

Table 1: *Average misclassification rate (%) of different learning objectives over 8 datasets. The best results are shown in bold with results that are not significant different to the best results underlined.*

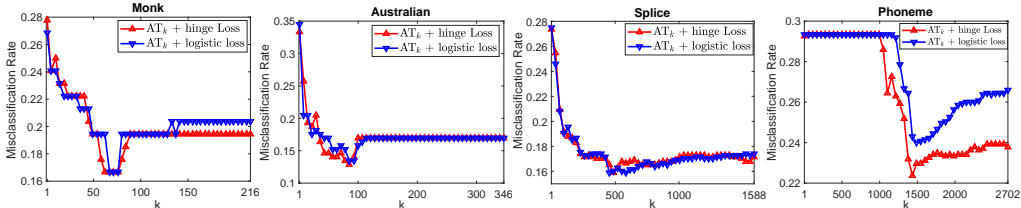

Figure 3: *Plots of misclassification rate on testing set vs. k on four datasets.*

We report the average performance over 10 random splitting of training/validation/testing for each dataset with $MAT_k$ learning objectives formed from individual logistic loss and hinge loss. Table 1 gives their experimental results in terms of misclassification rate (results in terms of other classification quality metrics are given in *supplementary materials*). Note that on these datasets, the average loss consistently outperforms the maximum loss, but the performance can be further improved with the $AT_k$ loss, which is more adaptable to different data distributions. This advantage of the $AT_k$ loss is particularly conspicuous for datasets Monk and Australian.

To further understand the behavior of $MAT_k$ learning on individual datasets, we show plots of misclassification rate on testing set vs. $k$ for four representative datasets in Fig.3 (in which $C$ is fixed to $10^2$ and $k \in [1, n]$). As these plots show, on all four datasets, there is a clear range of $k$ value with better classification performance than the two extreme cases $k = 1$ and $k = n$, corresponding to the maximum and average loss, respectively. To be more specific, when $k = 1$, the potential noises and outliers will have the highest negative effects on the learned classifier and the related classification performance is very poor. As $k$ increases, the negative effects of noises and outliers will reduce and the classification performance becomes better, this is more significant on dataset Monk, Australian and Splice. However, if $k$ keeps increase, the classification performance may decrease (*e.g.*, when $k = n$). This may because that as $k$ increases, more and more well classified samples will be included and the non-zero loss on these samples will have negative effects on the learned classifier (see our analysis in Section 2), specifically for dataset Monk, Australian and Phoneme.

**Regression.** Next, we report experimental results of linear regression on one synthetic dataset (Sinc) and three real datasets from [4], with a detailed description of these datasets given in *supplementary materials*. The standard square loss and absolute loss are adopted as individual losses. Note that average loss coupled with individual square loss is standard ridge regression model and average loss coupled with individual absolute loss reduces to $\nu$-SVR [19]. We normalize the target output to $[0, 1]$ and report their *root mean square error (RMSE)* in Table 2, with optimal $C$ and $k^*$ obtained by a grid search as in the case of classification (performance in terms of *mean absolute square error (MAE)* is given in supplementary materials). Similar to the classification cases, using the $AT_k$ loss usually improves performance in comparison to the average loss or maximum loss.

## 5  Related Works

Most work on learning objectives focus on designing individual losses, and only a few are dedicated to new forms of aggregate losses. Recently, aggregate loss considering the order of training data have been proposed in *curriculum learning* [2] and *self-paced learning* [11, 9], which suggest to organize the training process in several passes and samples are included from *easy* to *hard* gradually. It is interesting to note that each pass of *self-paced learning* [11] is equivalent to minimum the average of

|          | Square Loss | | | Absolute Loss | | |
|----------|-----------|-----------|-----------|-----------|-----------|-----------|
|          | Maximum | Average | AT$_{k^*}$ | Maximum | Average | AT$_{k^*}$ |
| Sinc     | 0.2790(0.0449) | 0.1147(0.0060) | **0.1139**(0.0057) | 0.1916(0.0771) | 0.1188(0.0067) | 0.1161(0.0060) |
| Housing  | 0.1531(0.0226) | 0.1065(0.0132) | **0.1050**(0.0132) | 0.1498(0.0125) | 0.1097(0.0180) | 0.1082(0.0189) |
| Abalone  | 0.1544(0.1012) | 0.0800(0.0026) | **0.0797**(0.0026) | 0.1243(0.0283) | 0.0814(0.0029) | 0.0811(0.0027) |
| Cpusmall | 0.2895(0.0722) | 0.1001(0.0035) | **0.0998**(0.0037) | 0.2041(0.0933) | 0.1170(0.0061) | 0.1164(0.0062) |

Table 2: *Average RMSE on four datasets. The best results are shown in bold with results that are not significant different to the best results underlined.*

the $k$ smallest individual losses, *i.e.*, $\frac{1}{k}\sum_{i=n-k+1}^{n}\ell_{[i]}(f)$, which we term it as the *average bottom-k* loss in contrast to the average top-k losses in our case. In [20], the pros and cons of the maximum loss and the average loss are compared, and the top-k loss, *i.e.*, $\ell_{[k]}(f)$, is advocated as a remedy to the problem of both. However, unlike the AT$_k$ loss, in general, neither the average bottom-k loss nor the top-k loss are convex functions with regards to the individual losses.

Minimizing top-$k$ errors has also been used in individual losses. For ranking problems, the work of [17, 24] describes a form of individual loss that gives more weights to the top examples in a ranked list. In multi-class classification, the top-1 loss is commonly used which causes penalties when the top-1 predicted class is not the same as the target class label [6]. This has been further extended in [12, 13] to the *top-k* multi-class loss, in which for a class label that can take $m$ different values, the classifier is only penalized when the correct value does not show up in the top $k$ most confident predicted values. As an individual loss, these works are complementary to the AT$_k$ loss and they can be combined to improve learning performance.

## 6 Discussion

In this work, we introduce the *average top-k* (AT$_k$) loss as a new aggregate loss for supervised learning, which is the average over the $k$ largest individual losses over a training dataset. We show that the AT$_k$ loss is a natural generalization of the two widely used aggregate losses, namely the average loss and the maximum loss, but can combine their advantages and mitigate their drawbacks to better adapt to different data distributions. We demonstrate that the AT$_k$ loss can better protect small subsets of hard samples from being swamped by a large number of easy ones, especially for imbalanced problems. Furthermore, it remains a convex function over all individual losses, which can lead to convex optimization problems that can be solved effectively with conventional gradient-based methods. We provide an intuitive interpretation of the AT$_k$ loss based on its equivalent effect on the continuous individual loss functions, suggesting that it can reduce the penalty on correctly classified data. We further study the theoretical aspects of AT$_k$ loss on classification calibration and error bounds of minimum average top-$k$ learning for AT$_k$-SVM. We demonstrate the applicability of minimum average top-$k$ learning for binary classification and regression using synthetic and real datasets.

There are many interesting questions left unanswered regarding using the AT$_k$ loss as learning objectives. Currently, we use conventional gradient-based algorithms for its optimization, but we are investigating special instantiations of MAT$_k$ learning for which more efficient optimization methods can be developed. Furthermore, the AT$_k$ loss can also be used for unsupervised learning problems (*e.g.*, clustering), which is a focus of our subsequent study. It is also of practical importance to combine AT$_k$ loss with other successful learning paradigms such as deep learning, and to apply it to large scale real life dataset. Lastly, it would be very interesting to derive error bounds of MAT$_k$ with general individual loss functions.

## 7 Acknowledgments

We thank the anonymous reviewers for their constructive comments. This work was completed when the first author was a visiting student at SUNY Albany, supported by a scholarship from University of Chinese Academy of Sciences (UCAS). Siwei Lyu is supported by the National Science Foundation (NSF, Grant IIS-1537257) and Yiming Ying is supported by the Simons Foundation (#422504) and the 2016-2017 Presidential Innovation Fund for Research and Scholarship (PIFRS) program from SUNY Albany. This work is also partially supported by the National Science Foundation of China (NSFC, Grant 61620106003) for Bao-Gang Hu and Yanbo Fan.

## Footnotes

[2]We define the *top-k* element of a set $S = \{s_1, \cdots, s_n\}$ as $s_{[k]}$, such that $s_{[1]} \ge s_{[2]} \ge \cdots \ge s_{[n]}$.

[3] https://archive.ics.uci.edu/ml/datasets.html

[4] http://sci2s.ugr.es/keel/datasets.php

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
