[Supplementary Material]

# Supplementary Materials for Learning with Average Top-k Loss

**Yanbo Fan**[3,4,1] , **Siwei Lyu**[1§], **Yiming Ying**[2] , **Bao-Gang Hu**[3,4]
[1]Department of Computer Science, University at Albany, SUNY
[2]Department of Mathematics and Statistics, University at Albany, SUNY
[3]National Laboratory of Pattern Recognition, CASIA
[4]University of Chinese Academy of Sciences (UCAS)
{yanbo.fan,hubg}@nlpr.ia.ac.cn, slyu@albany.edu, yying@albany.edu

## A Proofs

### A.1 Proofs of Lemma 1 and 2

**Proof of Lemma 1**

Notice that $\sum_{i=1}^{k} x_{[i]}$ is the solution of the following linear programming problem

$$\max_{\mathbf{p}} \ \mathbf{p}^T\mathbf{x}, \ \text{s.t.} \ \mathbf{p}^T\mathbf{1} = k, \mathbf{0} \leq \mathbf{p} \leq \mathbf{1}. \tag{8}$$

The Lagrangian of this linear programming problem is

$$L(\mathbf{p}, \mathbf{u}, \mathbf{v}, \lambda) = -\mathbf{p}^T\mathbf{x} - \mathbf{v}^T\mathbf{p} + \mathbf{u}^T(\mathbf{p} - \mathbf{1}) + \lambda(\mathbf{p}^T\mathbf{1} - k), \tag{9}$$

where $\mathbf{u} \geq 0$, $\mathbf{v} \geq 0$ and t are Lagrangian multipliers. Taking its derivative w.r.t $\mathbf{p}$ and set it to be $\mathbf{0}$, we have $\mathbf{v} = \mathbf{u} - \mathbf{x} + \lambda\mathbf{1}$. Substituting this back into the Lagrangian to eliminate the primal variable, we obtain the dual problem of (8) as

$$\min_{\mathbf{u},\lambda} \ \mathbf{u}^T\mathbf{1} + k\lambda, \ \text{s.t.} \ \mathbf{u} \geq \mathbf{0}, \mathbf{u} + \lambda\mathbf{1} - \mathbf{x} \geq \mathbf{0}. \tag{10}$$

This further means that

$$\sum_{i=1}^{k} x_{[i]} = \min_{\lambda}\Big\{k\lambda + \sum_{i=1}^{n} [x_i - \lambda]_{+}\Big\}. \tag{11}$$

The convexity of $\sum_{i=1}^{k} x_{[i]}$ follows directly from (11) and the fact that the partial minimum of a jointly convex function is convex. Furthermore, it is easy to see that $\lambda = x_{[k]}$ is always one optimal solution for (11), hence, for $x_i \geq 0, i = 1, \cdots, n$, there holds

$$\sum_{i=1}^{k} x_{[i]} = \min_{\lambda \geq 0}\Big\{k\lambda + \sum_{i=1}^{n} [x_i - \lambda]_{+}\Big\}. \tag{12}$$

$\square$

**Proof of Lemma 2**

Denote $g(\ell) = \big[[a - \ell]_{+} - b\big]_{+}$. For any $a \geq 0, b \geq 0$, we have $g(\ell) = 0 = [a - b - \ell]_{+}$ if $\ell \geq a$. In the Case of $\ell < a$, there holds $g(\ell) = [a - b - \ell]_{+}$. Thus $g(\ell) = [a - b - \ell]_{+}$ for any $a \geq 0, b \geq 0$.

$\square$

---

## A.2 Proof of Theorem 1

Note that $\ell : \mathbb{R} \to \mathbb{R}^+$ is convex, differentiable at 0 and $\ell'(0) < 0$ implies that $\ell(0) > 0$. Hence, by normalization we can let $\ell(0) = 1$. Indeed, the commonly used individual losses such as the least square loss $\ell(t) = (1-t)^2$, the hinge loss $\ell(t) = (1-t)_+$, and the logistic loss $\ell(t) = \log_2(1+e^{-t})$ satisfy the conditions $\ell'(0) < 0$ and $\ell(0) = 1$. The assumption in part (ii) of Theorem 1 implicitly assumes that $\mathcal{E}_\ell^* \leq 1$ because $\mathcal{E}_\ell^* \leq \mathcal{E}_\ell(0) = 1$.

Since $(f^*, \lambda^*)$ is a minimizer, then, by choosing $f = 0$ and $\lambda = \ell(0) = 1$ there holds $\mathbb{E}[\ell(yf^*(x)) - \lambda^*)_+] + \nu\lambda^* \leq \mathbb{E}[(\ell(0) - \ell(0))_+] + \nu\ell(0)$ which implies that the minimizer $\lambda^*$ defined in (7) must satisfy $0 \leq \lambda^* \leq \ell(0) = 1$. This means that the minimization over $\lambda$ in (7) can be restricted to $0 \leq \lambda \leq \ell(0) = 1$. Let $\beta = 1 - \lambda$ which implies that the minimization (7) is equivalent to the following

$$(f^*, \beta^*) = \arg\inf_{f, 0 \leq \beta \leq 1} \left\{ \mathbb{E}[(\beta + \ell(yf(\mathbf{x})) - 1)_+] - \nu\beta \right\}. \tag{13}$$

Let $(f^*, \beta^*)$ be the minimizer. We have, for any $f$ and choosing $\beta = \ell(0) = 1$, that

$$-\nu\beta^* \leq \left\{ \mathbb{E}[(\beta^* + \ell(yf^*(x)) - 1)_+] - \nu\beta^* \leq \mathbb{E}[(1 + \ell(yf(\mathbf{x})) - 1)_+] - \nu = \mathcal{E}_\ell(f) - \nu. \right.$$

This implies that $\nu\beta^* \geq \nu - \mathcal{E}_\ell(f)$. Since $f$ is arbitrary, $\beta^* \geq \frac{\nu - \mathcal{E}_\ell^*}{\nu} > 0$ if $\nu > \mathcal{E}_\ell^*$. Consequently, the above arguments show that $0 \leq \lambda^* = 1 - \beta^* < 1$ if $\nu > \mathcal{E}_\ell^*$.

Now observe that $f^* = \arg\inf_f \left\{ \mathbb{E}[(\ell(yf(\mathbf{x})) - \lambda^*)_+] + \nu\lambda^* \right\} = \arg\inf_f \left\{ \mathbb{E}[(\ell(yf(\mathbf{x})) - \lambda^*)_+] \right\}$. Define $\phi(t) = (\ell(t) - \lambda^*)_+$. This means that $f^* = \arg\inf_f \mathbb{E}[\phi(yf(\mathbf{x}))]$ for standard classification. The result of Theorem 2 in [1] states that that the loss $\phi$ is classification calibrated if $\phi$ is differentiable at 0 and $\phi'(0) < 0$. Notice that $\lambda^* < \ell(0) = 1$ as proved above, which implies that $\phi$ is differentiable at 0 and $\phi'(0) = \ell'(0) < 0$. This shows that $f^*$ has the same sign as the Bayes rule $\text{sign}(\Pr(y = 1|\mathbf{x}) - \frac{1}{2})$ if $\nu > \mathcal{E}_\ell^*$. This completes the proof of the first part of the theorem.

We now move on to prove the proof of the second part of the theorem. To this end, observe that $\lambda^* = \arg\inf_{\lambda \geq 0} \left\{ \mathbb{E}[\ell(yf^*(x)) - \lambda)_+] + \nu\lambda \right\}$. Assume that $\lambda^* = 0$. Then, $f^* = f_\ell^*$ and choosing $f = 0$ and $\lambda = 1 = \ell(0)$ in the objective function of (7) implies that $\nu = \mathbb{E}[(\ell(0) - 1)_+] + \nu \geq \mathbb{E}[\ell(yf^*(x)) - \lambda^*)_+] + \nu\lambda^* = \mathbb{E}[\ell(yf_\ell^*(x))] \geq \mathcal{R}^*$. Recall [8] that the Bayes error $\mathcal{R}^* = \int_{\mathcal{X}} \min(\eta(\mathbf{x}), 1 - \eta(\mathbf{x}))\rho_{\mathcal{X}}(\mathbf{x})$. This proves the Case $\lambda^* = 0$.

Now it only suffices to prove the Case of $\lambda^* > 0$. In this Case, by the first-order optimality condition, there exists a subgradient of $\mathbb{E}[\ell(yf^*(x)) - \lambda)_+] + \nu\lambda$ of the variable $\lambda$ at $\lambda^*$ equals to zero. This implies that $\mathbb{E}[h(x, y)] + \nu = 0$, where $h(x, y)$ is some subgradient of $(\ell(yf^*(x)) - \lambda)_+$ with respect to $\lambda$ at $\lambda^*$. Notice that $h(x, y) \leq -\mathbb{I}_{\ell(yf^*(x)) > \lambda^*}$. Consequently, $\nu \geq \mathbb{E}[\mathbb{I}_{\ell(yf^*(x)) > \lambda^*}] \geq \mathbb{E}[\mathbb{I}_{\ell(yf^*(x)) > \ell(0)}]$ since $\lambda^* \leq \ell(0)$ as proved in part (i). Since we assume that $\ell$ is monotonically decreasing, $\ell(yf^*(x)) > \ell(0)$ is equivalent to $yf^*(x) < 0$. The calibration of $\text{AT}_k$ models (i.e. $f^*$ has the sign as the Bayes rule) implies that $yf^*(x) < 0$ is equivalent to $y(2\eta(x)-1) < 0$. Putting the above arguments together, we conclude that $\nu \geq \mathbb{E}[\mathbb{I}_{y(2\eta(x)-1) < 0}] = \int_{\eta(x) \neq 1/2} \min(\eta(x), 1 - \eta(x))$. This completes the proof of the theorem.

## A.3 Proof of Theorem 2

Steinwart [22] derived the bounds for the excess misclassification error for $\nu$-SVM under the assumption that the kernel is *universal*, *i.e.*, the RKHS is dense in the space of continuous functions $\mathcal{C}(\mathcal{X})$ under the uniform norm $\| \cdot \|_\infty$ (See [23] for more details). The proof there depends on Urysohn's lemma in topology which states any two disjoint closed subsets can be separated by a continuous function. In contrast, our result holds true without the assumption of universal kernels.

To prove Theorem 2, we need some technical lemmas. We say the function $F : \prod_{k=1}^{m} \Omega_k \to \mathbb{R}$ has bounded differences $\{c_k\}_{k=1}^m$ if, for all $1 \leq k \leq m$,

$$\max_{z_1, \cdots, z_k, z_k' \cdots, z_m} |F(z_1, \cdots, z_{k-1}, z_k, z_{k+1}, \cdots, z_m) - F(z_1, \cdots, z_{k-1}, z_k', z_{k+1}, \cdots, z_m)| \leq c_k.$$

**Lemma 3.** *(McDiarmid's inequality [30]) Suppose $f : \prod_{k=1}^{m} \Omega_k \to \mathbb{R}$ has bounded differences $\{c_k\}_{k=1}^{m}$ then, for all $\epsilon > 0$, there holds*

$$\Pr\left\{ F(\mathbf{z}) - \mathbb{E}[F(\mathbf{z})] \geq \epsilon \right\} \leq e^{-\frac{2\epsilon^2}{\sum_{k=1}^{m} c_k^2}}.$$

We need to use the the Rademacher average and its contraction property [29, 31].

**Definition 1.** *Let $\mu$ be a probability measure on $\Omega$ and $F$ be a class of uniformly bounded functions. For every integer $m$, the Rademacher average over a set of functions $F$ on*

$$R_m(F) := \mathbb{E}_\mu \mathbb{E}_\epsilon \left\{ \frac{1}{m} \sup_{f \in F} \left| \sum_{i=1}^{m} \sigma_i f(z_i) \right| \right\}$$

*where $\{z_i\}_{i=1}^{m}$ are independent random variables distributed according to $\mu$ and $\{\sigma_i\}_{i=1}^{m}$ are independent Rademacher random variables, i.e., $\Pr(\sigma_i = +1) = \Pr(\sigma_i = -1) = 1/2$.*

**Lemma 4.** *Let $F$ be a class of uniformly bounded real-valued functions on $(\Omega, \mu)$ and $m \in \mathbb{N}$. If for each $i \in \{1, \ldots, m\}$, $\Psi_i : \mathbb{R} \to \mathbb{R}$ is a function with a Lipschitz constant $c_i$, then for any $\{x_i\}_{i=1}^{m}$,*

$$\mathbb{E}_\epsilon \left( \sup_{f \in F} \left| \sum_{i=1}^{m} \epsilon_i \Psi_i(f(x_i)) \right| \right) \leq 2\mathbb{E}_\epsilon \left( \sup_{f \in F} \left| \sum_{i=1}^{m} c_i \epsilon_i f(x_i) \right| \right). \tag{14}$$

Using the standard techniques involving Rademacher averages [29], one can get the following estimation. For completeness, we give a self-contained proof. Let the empirical error related to the hinge loss be denoted by $\mathcal{E}_{h,\mathbf{z}}(f) = \frac{1}{n} \sum_{i=1}^{n} (1 - yf(\mathbf{x}_i))_+$.

**Lemma 5.** *For any $\varepsilon > 0$, there holds*

$$\Pr\left\{ \sup_{\|f\|_K \leq R} \mathcal{E}_h(f) - \mathcal{E}_{h,\mathbf{z}}(f) \geq \varepsilon + \frac{2\kappa R}{\sqrt{n}} \right\} \leq e^{-\frac{2n\varepsilon^2}{(1+\kappa R)^2}}.$$

*Proof.* Let $F(z) = \sup_{\|f\|_K \leq R} [\mathcal{E}_h(f) - \mathcal{E}_{h,\mathbf{z}}(f)]$. Observe, for any $x, y$, that $(1 - yf(x))_+ \leq 1 + |f(x)| \leq 1 + |\langle K_x, f \rangle_K| \leq 1 + \|f\| \langle K_x, K_x \rangle_K^{\frac{1}{2}} = 1 + \|f\|_K \sqrt{K(x,x)} \leq \kappa R$. Then, one can easily get that the bounded differences are $c_k = \frac{1+\kappa R}{n}$ for any $1 \leq k \leq n$. By the McDiarmid inequality, we have

$$\Pr\left\{ \sup_{\|f\|_K \leq R} [\mathcal{E}_h(f) - \mathcal{E}_{h,\mathbf{z}}(f)] \geq \mathbb{E}_\mathbf{z} \sup_{\|f\|_K \leq R} [\mathcal{E}_h(f) - \mathcal{E}_{h,\mathbf{z}}(f)] + \varepsilon \right\} \geq \exp\left\{ -\frac{2n\varepsilon^2}{(1+\kappa R)^2} \right\}.$$

Let $\mathbf{z}' = \{z_1', z_2', \ldots, z_n'\}$ be i.i.d. copies of $\mathbf{z}$. Then,

$$\mathbb{E}_\mathbf{z} \sup_{\|f\|_K \leq R} [\mathcal{E}_h(f) - \mathcal{E}_{h,\mathbf{z}}(f)] = \mathbb{E}_\mathbf{z}[\sup_{\|f\|_K \leq R} [\mathbb{E}_{\mathbf{z}'}(\mathcal{E}_{\mathbf{z}'}(f)) - \mathcal{E}_{h,\mathbf{z}}(f)] \leq \mathbb{E}_\mathbf{z}\mathbb{E}_{\mathbf{z}'} \sup_{\|f\|_K \leq R} [\mathcal{E}_{\mathbf{z}'}(f) - \mathcal{E}_{h,\mathbf{z}}(f)].$$

By standard symmetrization techniques [29], for any Rademacher variables $\{\sigma_i : i = 1, \ldots, n\}$, we have that

$$\mathbb{E}_\mathbf{z}\mathbb{E}_{\mathbf{z}'} \sup_{\|f\|_K \leq R} [\mathcal{E}_h(f) - \mathcal{E}_{h,\mathbf{z}}(f)] = \mathbb{E}_\mathbf{z}\mathbb{E}_{\mathbf{z}'}\mathbb{E}_\sigma \sup_{\|f\|_K \leq R} [\frac{1}{n} \sum_{i=1}^{n} \sigma_i((1 - y_i'f(x_i'))_+ - (1 - y_if(x_i))_+)]$$

$$= 2\mathbb{E}_\mathbf{z}\mathbb{E}_\sigma \sup_{\|f\|_K \leq R} [\frac{1}{n} \sum_{i=1}^{n} \sigma_i(1 - y_if(x_i))_+] \leq 2\mathbb{E}_\mathbf{z}\mathbb{E}_\sigma \sup_{\|f\|_K \leq R} \frac{1}{n} \left| \sum_{i=1}^{n} \sigma_i(1 - y_if(x_i))_+ \right|.$$

Let $\Phi_i(t) = (1 - y_it)_+$ which has Lipschitz constant 1. By the contraction property of Rademacher averages,

$$\mathbb{E}_\sigma \sup_{\|f\|_K \leq R} \frac{1}{n} \left| \sum_{i=1}^{n} \sigma_i(1 - y_if(x_i))_+ \right| \leq \mathbb{E}_\sigma \sup_{\|f\|_K \leq R} \frac{1}{n} \left| \sum_{i=1}^{n} \sigma_i f(x_i) \right| = \mathbb{E}_\sigma \sup_{\|f\|_K \leq R} \left| \langle \frac{1}{n} \sum_{i=1}^{n} \sigma_i K_{x_i}, f \rangle \right|$$

$$\leq \mathbb{E}_\sigma \sup_{\|f\|_K \leq R} \|\frac{1}{n}\sum_{i=1}^n \sigma_i K_{x_i}\|_K \|f\|_K = R\,\mathbb{E}_\sigma\Big[\|\frac{1}{n}\sum_{i=1}^n \sigma_i K_{x_i}\|_K\Big]$$

$$\leq R\Big[\mathbb{E}_\sigma\|\frac{1}{n}\sum_{i=1}^n \sigma_i K_{x_i}\|_K^2\Big]^{\frac{1}{2}} \leq \frac{R}{n}\Big[\sum_{i=1}^n K(x_i,x_i)\Big]^{\frac{1}{2}} \leq \frac{\kappa R}{\sqrt{n}}.$$

Putting all the above estimations together yields the desired result. This completes the proof of the lemma. $\qquad\square$

We also need the Höeffding's inequality stated as follows.

**Lemma 6.** *Let $\xi$ be a random variable and, for any $i \in [m]$, $a_i \leq \xi \leq b_i$. Then, for any $\varepsilon > 0$, there holds*

$$\Pr\Big\{\frac{1}{m}\sum_{i=1}^m \xi_i - E\xi \geq \epsilon\Big\} \leq \exp\Big\{-\frac{m\epsilon^2}{2M^2}\Big\}.$$

To prove the main theorem, we need to establish a lower bound for $\rho_\mathbf{z}$. Denote $\kappa = \sup_{x \in \mathcal{X}}\sqrt{K(x,x)}$.

**Lemma 7.** *For $\mu \in (0, 1 - \mathcal{E}_h(f_\mathcal{H}))$, let $\lceil n(\mathcal{E}_h(f_\mathcal{H}) + \mu)\rceil \leq k \leq n$, then we have*

$$\Pr\Big\{\mathbf{z} \in \mathcal{Z}^n : \frac{\|f_\mathbf{z}\|_K}{\rho_\mathbf{z}} \leq \frac{2k}{n}\max\Big(\sqrt{\frac{2C}{\mu}}, \frac{2\|f_\mathcal{H}\|_K}{\mu}\Big)\Big\} \geq 1 - \exp\Big\{-\frac{n\mu^2}{2(1+\kappa\|f_\mathcal{H}\|_K)^2}\Big\}.$$

*Proof.* Since $(f_\mathbf{z}, \rho_\mathbf{z})$ is a minimizer of formulation (6), for any $0 < \rho \leq 1$ there holds

$$\frac{1}{n}\sum_{i=1}^n (\rho_\mathbf{z} - y_i f_\mathbf{z}(x_i))_+ - \frac{k}{n}\rho_\mathbf{z} + \frac{1}{2C}\|f_\mathbf{z}\|_K^2 \leq \frac{1}{n}\sum_{i=1}^n (\rho - y_i \rho f_\mathcal{H}(x_i))_+ - \frac{k}{n}\rho + \frac{1}{2C}\|\rho f_\mathcal{H}\|_K^2$$

$$= \rho\mathcal{E}_{h,\mathbf{z}}(f_\mathcal{H}) - \frac{k}{n}\rho + \frac{\rho^2}{2C}\|f_\mathcal{H}\|_K^2. \qquad (15)$$

This implies, for any $0 < \rho \leq 1$, that

$$\frac{k}{n}\rho_\mathbf{z} \geq -\rho\mathcal{E}_{h,\mathbf{z}}(f_\mathcal{H}) + \frac{k}{n}\rho - \frac{\rho^2}{2C}\|f_\mathcal{H}\|_K^2.$$

Applying the Hoeffding inequality (Lemma 6) yields that

$$\Pr\Big\{\mathcal{E}_{h,\mathbf{z}}(f_\mathcal{H}) - \mathcal{E}_h(f_\mathcal{H}) \leq \frac{\mu}{2}\Big\} \leq 1 - \exp\Big\{-\frac{n\mu^2}{2(1+\kappa\|f_\mathcal{H}\|_K)^2}\Big\}. \qquad (16)$$

Then, on the event $\mathcal{U} = \{\mathbf{z} \in \mathcal{Z}^n : \mathcal{E}_{h,\mathbf{z}}(f_\mathcal{H}) - \mathcal{E}_h(f_\mathcal{H}) \leq \frac{\mu}{2}\}$, we have $-\rho\mathcal{E}_{h,\mathbf{z}}(f_\mathcal{H}) + \frac{k}{n}\rho - \frac{\rho^2}{2C}\|f_\mathcal{H}\|_K^2 \geq \rho(\frac{k}{n} - \mathcal{E}(f_\mathcal{H}) - \frac{\mu}{2}) - \frac{\rho^2}{2C}\|f_\mathcal{H}\|_K^2 \geq \rho\frac{\mu}{2} - \frac{\rho^2}{2C}\|f_\mathcal{H}\|_K^2$. Define $g(\rho) = \frac{\rho\mu}{2} - \frac{\rho^2}{2C}\|f_\mathcal{H}\|_K^2$. It is easy to observe that

$$\max_{0 < \rho \leq 1} g(\rho) \geq \begin{cases} \frac{C\mu^2}{8\|f_\mathcal{H}\|_K^2}, & C\mu \leq 2\|f_\mathcal{H}\|_K^2, \\ \frac{\mu}{4}, & C\mu > 2\|f_\mathcal{H}\|_K^2. \end{cases}$$

Consequently, on the event $\mathcal{U}$, there holds

$$\rho_\mathbf{z} \geq \frac{n}{k}\max_{0 < \rho \leq 1} g(\rho) \geq \frac{n}{k}\min\Big(\frac{\mu}{4}, \frac{C\mu^2}{8\|f_\mathcal{H}\|_K^2}\Big). \qquad (17)$$

By choosing $\rho = 0$ in (15), there holds $\frac{\|f_\mathbf{z}\|_K^2}{\rho_\mathbf{z}} \leq \frac{2Ck}{n}$. Combining these estimation together, on the event $\mathcal{U}$ there holds

$$\frac{\|f_\mathbf{z}\|_K}{\rho_\mathbf{z}} \leq \sqrt{\frac{\|f_\mathbf{z}\|_K^2}{\rho_\mathbf{z}}}\sqrt{\frac{1}{\rho_\mathbf{z}}} \leq \frac{2k}{n}\max\Big(\sqrt{\frac{2C}{\mu}}, \frac{2\|f_\mathcal{H}\|_K}{\mu}\Big).$$

This completes the proof of the lemma. $\qquad\square$

With all the above technical lemmas, we are ready to prove Theorem 2.

**Proof of Theorem 2.** We will use the relationship between the excess misclassification error and generalization error [32], *i.e.* for any $f : \mathcal{X} \to \mathbb{R}$, there holds

$$\mathcal{R}(\mathrm{sign}(f)) - \mathcal{R}(f_c) \leq \mathcal{E}_h(f) - \mathcal{E}_h(f_c). \tag{18}$$

Let $\mathcal{U}_1$ be the event such that the inequality in Lemma 7 is true, *i.e.* $\mathcal{U}_1 = \left\{ \mathbf{z} \in \mathcal{Z}^n : \frac{\|f_{\mathbf{z}}\|_K}{\rho_{\mathbf{z}}} \leq \frac{2k}{n} \max\left( \sqrt{\frac{2C}{\mu}}, \frac{2\|f_{\mathcal{H}}\|_K}{\mu} \right) \right\}$. On the event $\mathcal{U}_1$, noting that $0 < \mu \leq 1$ we have that $\frac{\|f_{\mathbf{z}}\|_K}{\rho_{\mathbf{z}}} \leq R_{C,\mu} := \frac{2\sqrt{2C}+4\|f_{\mathcal{H}}\|_K}{\mu}$.

Now considering the sample $\mathbf{z} \in \mathcal{U}_1$, using (18) we have

$$\mathcal{R}(\mathrm{sign}(f_{\mathbf{z}})) - \mathcal{R}(f_c) \leq \mathcal{E}_h\big(\frac{f_{\mathbf{z}}}{\rho_{\mathbf{z}}}\big) - \mathcal{E}(f_c) \leq \mathcal{E}_h\big(\frac{f_{\mathbf{z}}}{\rho_{\mathbf{z}}}\big) - \mathcal{E}_{h,\mathbf{z}}\big(\frac{f_{\mathbf{z}}}{\rho_{\mathbf{z}}}\big) + \mathcal{E}_{h,\mathbf{z}}\big(\frac{f_{\mathbf{z}}}{\rho_{\mathbf{z}}}\big) - \mathcal{E}(f_c) \tag{19}$$

By the definition of the minimizer $(\rho_{\mathbf{z}}, f_{\mathbf{z}})$, there holds $\frac{1}{n}\sum_{i=1}^{n}(\rho_{\mathbf{z}} - y_i f_{\mathbf{z}}(x_i))_+ - \frac{k}{n}\rho_{\mathbf{z}} + \frac{1}{2C}\|f_{\mathbf{z}}\|_K^2 \leq 0$ which means that $\frac{1}{n}\sum_{i=1}^{n}(\rho_{\mathbf{z}} - y_i f_{\mathbf{z}}(x_i))_+ \leq \frac{k}{n}\rho_{\mathbf{z}}$. Equivalently, $\mathcal{E}_{h,\mathbf{z}}\big(\frac{f_{\mathbf{z}}}{\rho_{\mathbf{z}}}\big) \leq \frac{k}{n}$ on the event $\mathcal{U}_1$. This combines with (19) implies, on the event $\mathcal{U}_1$, that

$$\mathcal{R}(\mathrm{sign}(f_{\mathbf{z}})) - \mathcal{R}(f_c) \leq \mathcal{E}_h\big(\frac{f_{\mathbf{z}}}{\rho_{\mathbf{z}}}\big) - \mathcal{E}_{h,\mathbf{z}}\big(\frac{f_{\mathbf{z}}}{\rho_{\mathbf{z}}}\big) + \big(\frac{k}{n} - \mathcal{E}_h(f_{\mathcal{H}})\big) + \mathcal{E}_h(f_{\mathcal{H}}) - \mathcal{E}(f_c)$$

$$\leq \sup_{\|f\|_K \leq R_{C,\mu}} \Big[ \mathcal{E}_h(f) - \mathcal{E}_{h,\mathbf{z}}(f) \Big] + \big(\frac{k}{n} - \mathcal{E}_h(f_{\mathcal{H}})\big) + \inf_{f \in \mathcal{H}_K} \mathcal{E}_h(f) - \mathcal{E}_h(f_c)$$

$$\leq \sup_{\|f\|_K \leq R_{C,\mu}} \Big[ \mathcal{E}_h(f) - \mathcal{E}_{h,\mathbf{z}}(f) \Big] + \big(\frac{k}{n} - \mathcal{E}_h(f_{\mathcal{H}})\big) + \mathcal{A}(\mathcal{H}_K)$$

$$\leq \sup_{\|f\|_K \leq R_{C,\mu}} \Big[ \mathcal{E}_h(f) - \mathcal{E}_{h,\mathbf{z}}(f) \Big] + \mu + \frac{1}{n} + \mathcal{A}(\mathcal{H}_K),$$

where the last inequality follows from the fact, by the definition $k = k(n) = \lceil n(\mathcal{E}_h(f_{\mathcal{H}}) + \mu) \rceil$, that $\mathcal{E}_h(f_{\mathcal{H}}) + \mu \leq \frac{k}{n} \leq \mathcal{E}_h(f_{\mathcal{H}}) + \mu + \frac{1}{n}$. Therefore,

$$\Pr\bigg\{ \mathbf{z} \in \mathcal{Z}^n : \mathcal{R}(\mathrm{sign}(f_{\mathbf{z}})) - \mathcal{R}(f_c) \geq \mu + \frac{1}{n} + \mathcal{A}(\mathcal{H}) + \varepsilon + \frac{2\kappa R_{C,\mu}}{\sqrt{n}} \bigg\}$$

$$\leq \Pr(\mathcal{U}_1^c) + \Pr\bigg\{ \mathbf{z} \in \mathcal{U}_1 : \sup_{\|f\|_K \leq R_{C,\mu}} \big[ \mathcal{E}_h(f) - \mathcal{E}_{h,\mathbf{z}}(f) \big] \geq \varepsilon + \frac{2\kappa R_{C,\mu}}{\sqrt{n}} \bigg\}$$

$$\leq \exp\bigg(-\frac{n\mu^2}{2(1+\kappa\|f_{\mathcal{H}}\|_K)^2}\bigg) + \Pr\bigg\{ \mathbf{z} : \sup_{\|f\|_K \leq R_{C,\mu}} \big[ \mathcal{E}_h(f) - \mathcal{E}_{h,\mathbf{z}}(f) \big] \geq \varepsilon + \frac{2\kappa R_{C,\mu}}{\sqrt{n}} \bigg\}$$

$$\leq \exp\bigg(-\frac{n\mu^2}{2(1+\kappa\|f_{\mathcal{H}}\|_K)^2}\bigg) + \exp\bigg(-\frac{2n\varepsilon^2}{(1+\kappa R_{C,\mu})^2}\bigg)$$

$$\leq 2\exp\bigg(-\frac{n\varepsilon^2\mu^2}{(1+2\kappa\sqrt{2C}+4\kappa\|f_{\mathcal{H}}\|_K)^2}\bigg).$$

Here, the second to last inequality follows from Lemma 5 which is the standard estimation for Rademacher averages [29]. $\qquad\square$

## B  Examples of $\mathrm{AT}_k$ loss coupled with different individual losses

The proposed $\mathrm{AT}_k$ loss is quite general and can be combined with different existing individual losses. An interesting phenomenon is that $\mathrm{AT}_k$ with hinge loss and absolute loss have a close relations to the well-known $\nu$-SVM and $\nu$-SVR that proposed in [19], respectively. Specifically, we have

**Proposition 1.** *Under conditions* $C = 1$ *and* $K(\mathbf{x}_i, \mathbf{x}_i) \leq 1$ *for any* $i$, $\mathrm{AT}_k$*-SVM* (6) *reduces to* $\nu$*-SVM with* $\nu = \frac{k}{n}$.

*Proof.* Recall [19] that the primal problem of the $\nu$-SVM without the bias term $b$ is formulated by

$$\min_{f\in\mathcal{H}_K,\rho\geq 0} \frac{1}{n}\sum_{i=1}^{n}[\rho-y_if(\mathbf{x}_i)]_+ - \nu\rho + \frac{1}{2}\|f\|_K^2, \tag{20}$$

where $\nu\in[0,1]$ is a scalar. Its dual is given by

$$\begin{cases} \min_\alpha & \frac{1}{2}\sum_{i,j=1}^{n}\alpha_i\alpha_jy_iy_jK(x_i,x_j) \\ \text{s.t.} & 0\leq\alpha_i\leq\frac{1}{n},\forall i=1,2,\ldots,n \\ & \sum_{i=1}^{n}\alpha_i\geq\nu. \end{cases}$$

The KKT conditions implies, for any optimal solution $\alpha^*$ of the dual and any optimal solution $(f_{\mathbf{z}},\rho_{\mathbf{z}})$ of the primal, there holds, for the support vectors $\mathbf{x}_i$ with $0 < \alpha_i^* < \frac{1}{n}$, that $\rho_{\mathbf{z}} = y_i\sum_{j=1}^{n}\alpha_j^*y_jK(\mathbf{x}_i,\mathbf{x}_j)$. If one assumes that $K(\mathbf{x}_i,\mathbf{x}_i)\leq 1$ for all $i$, then $|K(x_i,x_j)| = |\langle K_{\mathbf{x}_i},K_{\mathbf{x}_j}\rangle_K| \leq \sqrt{K(\mathbf{x}_i,\mathbf{x}_i)}\sqrt{K(\mathbf{x}_j,\mathbf{x}_j)}\leq 1$. Therefore,

$$\rho_{\mathbf{z}} \leq |y_i\sum_{j=1}^{n}\alpha_j^*y_jK(\mathbf{x}_i,\mathbf{x}_j)| \leq \sum_{j=1}^{n}\alpha_j^* \leq 1,$$

where the last inequality follows from the fact that $\alpha_j\leq\frac{1}{n}$ for all $j$. Consequently, in the minimization of (20) we can restrict to $\rho\leq 1$ which implies that the $\text{AT}_k$-SVM (6) with $C=1$ is reduced to $\nu$-SVM with $\nu=\frac{k}{n}$. □

Besides, the dual formulation of $\text{AT}_k$-SVM (6) can be easily derived as

$$\begin{cases} \min_\alpha & \frac{1}{2}\sum_{i,j=1}^{n}\alpha_i\alpha_jy_iy_jK(x_i,x_j) - \sum_{i=1}^{n}\alpha_i \\ \text{s.t.} & 0\leq\alpha_i\leq\frac{C}{n},\forall i=1,2,\ldots,n \\ & \sum_{i=1}^{n}\alpha_i\leq\frac{Ck}{n}. \end{cases}$$

This leads to a convex quadratic programming problem for $\text{AT}_k$-SVM and can be solved efficiently.

**Proposition 2.** *$MAT_k$ model (3) coupled with absolute loss in the RKHS setting becomes $\nu$-SVR with $\nu=\frac{k}{n}$.*

*Proof.* Recall [19] that the primal problem of the $\nu$-SVR without the bias term $b$ in RKHS is formulated by

$$\min_{\mathbf{w},\lambda\geq 0} \frac{1}{n}\sum_{i=1}^{n}[|y_i-f(x_i)|-\lambda]_+ + \nu\lambda + \frac{1}{2C}\|f\|_K^2, \tag{21}$$

where $\nu\in[0,1]$ is a scalar. It is easy to see in the setting of RKHS that, with individual absolute loss (*i.e.*, $\ell(f(\mathbf{x}_i),y_i)=|y_i-f(\mathbf{x}_i)|$) and $\Omega(\mathbf{w})=\frac{1}{2C}\|f\|_K^2$, $\text{MAT}_k$ model (3) becomes

$$\min_{\mathbf{w},\lambda\geq 0} \frac{1}{n}\sum_{i=1}^{n}[|y_i-f(\mathbf{x}_i)|-\lambda]_+ + \frac{k}{n}\lambda + \frac{1}{2C}\|f\|_K^2, \tag{22}$$

We name model (22) as $\text{AT}_k$-SVR for brevity. It is straightforward that $\text{AT}_k$-SVR is exactly the $\nu$-SVR with $\nu=\frac{k}{n}$.

The above propositions provide new perspectives to understand the success of $\nu$-SVM and $\nu$-SVR. That is, through "shifting down" the original individual hinge loss and absolute loss and truncating them at zero, the penalty of correctly classified samples that are "far enough" from classification boundary in classification and the penalty of samples that are "close enough" to the regression tube in regression will be zero, which enables the model to put more effort to misclassified samples or samples that are "too far" to the regression tube. Besides, the good properties of $\nu$ in $\nu$-SVM and

| Binary Classification | | | | | | | | | | Regression | | |
|---|---|---|---|---|---|---|---|---|---|---|---|---|
| Dataset | $c$ | $n$ | $d$ | $IR$ | Dataset | $c$ | $n$ | $d$ | $IR$ | Dataset | $n$ | $d$ |
| Monk | 2 | 432 | 6 | 1.12 | Spambase | 2 | 4601 | 57 | 1.54 | Sinc | 1000 | 10 |
| Australian | 2 | 690 | 14 | 1.25 | German | 2 | 1000 | 24 | 2.33 | Housing | 506 | 13 |
| Madelon | 2 | 2600 | 500 | 1.0 | Titanic | 2 | 2201 | 3 | 2.10 | Abalone | 4177 | 8 |
| Splice | 2 | 3175 | 60 | 1.08 | Phoneme | 2 | 5404 | 5 | 2.41 | Cpusmall | 8192 | 12 |

Table 3: *Statistical information of each dataset, where $c, n, d$ are the number of classes, samples and features, respectively.* IR *is the class ratio.*

$\nu$-SVR that derived in [19] can be extended to $k$ in $\text{AT}_k$-SVM and $\text{AT}_k$-SVR directly. For example, for $\text{AT}_k$-SVM with conditions $C = 1$ and $K(\mathbf{x}_i, \mathbf{x}_i) \leq 1$ and $\text{AT}_k$-SVR, $k$ is a lower bound on the number of support vectors and is an upper bound on the number of margin errors. Due to its directness, we refer to [19] for their proofs. This can also help us select $k$ in $\text{AT}_k$-SVM and $\text{AT}_k$-SVR.

## C  Toy examples for effects of different aggregate losses

We illustrate the behaviors of different aggregate losses using binary classification on 2D synthetic data examples. We generate six different datasets (Fig. 4). Each dataset consists of 200 samples sampled from Gaussian distributions with distinct centers and variances. We use linear classifier and consider different aggregate losses combined with individual logistic loss and individual hinge loss. The learned linear classifiers and the misclassification rate of $\text{AT}_k$ vs. $k$ are shown in Fig. 4. The left panel in Fig. 4 (*i.e.*, (a1-a6) and (b1-b6)) refers to the results of aggregate losses combined with individual logistic loss and the right panel (*i.e.*, (c1-c6) and (d1-d6)) refers to the results of aggregate losses combined with individual hinge loss.

**Case 1.** The first row in Fig. 4 represents an ideal situation where there is no outliers and the $+$ samples and $-$ samples are well distributed and linear separable. In this Case , all aggregate losses with both logistic loss and hinge loss can get perfect classification results. This is also verified in Fig. 4 (b1) and Fig. 4 (d1) that the misclassification rate is zero for $\text{AT}_k$ with all $k$.

**Case 2.** In the second row, there exists an outlier in the $+$ class (shown as an enlarged $\times$). We can see that the maximum loss is very sensitive to outliers and its classification boundary in Fig. 4 (a2) and Fig. 4 (c2) are largely influenced by this outlier. Seen from Fig. 4 (b2) and Fig. 4 (d2), $\text{AT}_k$ loss is more robust with larger $k$ and achieves better classification results when $k \geq 3$.

**Case 3.** In the third row, there is no outliers and the $+$ samples and $-$ samples are still linear separable. However, the $+$ samples clearly has two distributions (typical distribution and rare distribution). Seen from Fig. 4 (a3) and Fig. 4 (c3), the linear classifiers learned from average loss sacrifice some $+$ samples from rare distribution even though the data are separable. This is because that the individual logistic loss has non-zero penalty for correctly classified samples and individual hinge loss has non-zero penalty for correctly classified samples with margin less than 1. Hence samples that are "too close" to the classification boundary (samples from rare distributions in this example) are sacrificed to accommodate reducing the average loss over the whole datasets. Besides, average with hinge loss achieves better results than that with logistic loss, this may because that for correctly classified samples with margin larger than 1, the penalty caused by hinge loss is zero while that caused by logistic loss is still non-zeros. Hence this part of samples still has "negative" effect to the learned classification boundary of average with logistic loss. By "shifting down" and truncating, $\text{AT}_k$ loss with proper $k$ (*e.g.*, $k \in [1, 18]$ for logistic loss and $k \in [1, 50]$ for hinge loss) can better fit this data, as is shown in Fig. 4 (b3) and Fig. 4 (d3).

**Case 4.** The plots in the fourth row refers to a more complicated situation where there are both multi-modal distributions and outliers. Obviously, neither maximum loss (due to the outlier) nor average loss (due to the multi-modal distributions) can fit this data very well. Seen from Fig. 4 (b4) and Fig. 4 (d4), there exists a proper region of $k$ (*i.e.*, $k \in [4, 24]$ for logistic loss and $k \in [3, 62]$ for hinge loss) that can yield much better classification results. We also report the linear classifier learned from $\text{AT}_{k=10}$ for better understanding. Seen from Fig. 4 (a4) and Fig. 4 (c4), the classification boundary of $\text{AT}_{k=10}$ is closer to the optimal Bayes linear classifier than that of maximum and average.

**Case 5.** The fifth row shows an imbalance scenario where the $-$ samples are far less that the $+$ ones. The $+$ samples and $-$ samples are linear separable. We can see from Fig. 4 (a5) that the average loss with individual logistic loss sacrifices all $-$ samples to obtain a small loss over the whole dataset.

|  | Logistic Loss | | | Hinge Loss | | |
|---|---|---|---|---|---|---|
|  | Maximum | Average | $\text{AT}_{k*}$ | Maximum | Average | $\text{AT}_{k*}$ |
| Monk | 75.80(3.37) | 79.47(2.05) | **82.95(2.39)** | 76.37(3.51) | 81.15(3.11) | 82.68(2.79) |
| Australian | 78.88(7.56) | 86.10(3.19) | **88.37(2.97)** | 78.99(7.47) | 85.72(3.15) | 87.50(4.14) |
| Madelon | 51.20(2.92) | 59.28(1.41) | **60.26(1.58)** | 49.42(2.71) | 59.36(1.83) | 59.72(1.51) |
| Splice | 76.31(1.94) | 82.73(1.01) | **83.90(0.97)** | 76.47(2.12) | 83.78(1.12) | 83.79(0.97) |
| Spambase | 69.48(5.94) | 90.63(1.21) | 90.63(1.21) | 69.96(6.76) | **91.90(0.85)** | **91.90(0.85)** |
| German | 44.88(7.37) | 60.12(7.59) | **63.80(4.29)** | 44.87(7.34) | 61.02(7.49) | 62.96(3.33) |
| Titanic | 46.52(15.27) | 66.69(1.44) | 66.69(1.44) | 48.55(13.15) | 66.65(1.43) | **67.74(1.78)** |
| Phoneme | 19.10(11.84) | 63.00(1.84) | 66.29(2.04) | 12.89(11.47) | **70.41(1.65)** | **70.41(1.65)** |

Table 4: *Average G-mean(%) of different learning objectives over* 8 *datasets. The best results are shown in bold with results that are not significant different to the best results underlined.*

|  | Square Loss | | | Absolute Loss | | |
|---|---|---|---|---|---|---|
|  | Maximum | Average | $\text{AT}_{k*}$ | Maximum | Average | $\text{AT}_{k*}$ |
| Sinc | 0.2438(0.0445) | 0.0816(0.0045) | **0.0806**(0.0044) | 0.1489(0.0466) | 0.0827(0.0048) | 0.0821(0.0055) |
| Housing | 0.1198(0.0150) | 0.0738(0.0075) | 0.0736(0.0079) | 0.1233(0.0127) | 0.0713(0.0089) | **0.0712**(0.0088) |
| Abalone | 0.1312(0.0919) | 0.0575(0.0016) | 0.0574(0.0015) | 0.1082(0.0303) | 0.0559(0.0014) | **0.0557**(0.0016) |
| Cpusmall | 0.2404(0.0832) | 0.0634(0.0027) | 0.0627(0.0025) | 0.1868(0.0997) | 0.0423(0.0018) | **0.0422**(0.0018) |

Table 5: *Average MAE on four datasets. The best results are shown in bold with results that are not significant different to the best results underlined.*

While the average loss with individual hinge loss obtains better results, it still sacrifices half of the $-$ samples, as is shown in Fig. 4 (c5). In contrast, $\text{AT}_k$ loss can better fit this distributions and achieves better classification results with $k \in [1, 25]$ for logistic loss and $k \in [1, 135]$ for hinge loss.

**Case 6.** The sixth row shows an imbalanced data with one outlier. Comparing to the results in the fifth row, the performance of maximum loss decreases due to the outlier. The performance of average loss with hinge loss also decreases. Seen from Fig. 4 (b6) and Fig. 4 (d6), $\text{AT}_k$ loss with $k \in [2, 12]$ for logistic loss and $k \in [3, 59]$ for hinge loss can better fit this data and achieve better classification results.

Though very simple, these synthetic datasets reveal some properties of the maximum loss and average loss intuitively. That is, while maximum loss performs very well for separable data, it is very sensitive to outliers. Meanwhile, average loss is more robust to outliers than maximum loss, however, it may sacrifices some correctly classified samples that are "too close" to the classification boundary, especially in imbalanced or multi-modal data distributions. As the distributions of datasets from real applications can be very complicated and outliers are unavoidable, it is interesting and helpful to add an extra freedom $k$ to better fitting different data distributions.

**Sinc data used for regression:** This dataset is drawn from sinc function, *i.e.*, $y = \sin(x)/x$, where $x$ is an scalar, and the goal is to estimate $y$ from the input $x$. We randomly select 1000 samples $(x_i, y_i)$ with $x_i$ drawn uniformly from $[-10, 10]$. As we use linear regression model in our experiments, we map the input $x$ into a kernel space via the radial basis function (RBF) kernel. We select 10 RBF kernels from $[-10, 10]$, which leads to 10-dimension input $\mathbf{x} = [k(x, c_1), \cdots, k(x, c_{10})]^T$, where $k(x, c_i) = \exp(-(x - c_i)^2)$. We also add random Gaussian noise $N(0, 0.2^2)$ to the target output.

Table 3 tabulates the statistical information of datasets that used in this paper. Experiments results in terms of G-mean for binary classification are reported in Table 4, and experiments results in terms of *mean absolute error (MAE)* for regression are also reported in Table 5.

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

Figure 4: *Comparison of different aggregate losses on 2D synthetic datasets for binary classification on six different data distributions. Each row refers to one data distribution. In all plots, the + samples are red crosses and the − samples are blue circles. The outliers are shown with an enlarged × if any. The plots on the left panel report the results of linear classifiers learned with different aggregate losses combined with individual logistic loss, and that on the right panel are the results of different aggregate losses combined with individual hinge loss. The plots on the first and third columns show the learned linear classifiers of maximum, average and $AT_{k=10}$ with the optimal Bayes classification shown as shaded areas, and the plots on the second and fourth columns show the misclassification rate of $AT_k$ vs. k.*