[Reviews · NeurIPS 2017]

Reviewer 1



This is an interesting paper that introduces and analyses a new way of aggregating individual losses over training examples, being an alternative for the commonly used average loss and the recently introduced maximum loss. The proposed average top-k loss lies in between of those two existing approaches. The premises concerning an alternative to the average loss shown in the beginning of the paper seem to be very valid. Indeed the behavior of the average loss in the situations presented in Figure 1 and described in the corresponding paragraphs are a good justification for this research topic. Also the analysis of the behavior of the maximum loss in comparison with average and average top-k for the case of non-separable data is interesting. Interestingly, in the experiments on all the datasets the accuracy of the model optimized for average loss is better than the one optimized for max loss. According to [19] there are data sets for which max loss should perform better than the average loss. It would be interesting to include such data sets to the experimental study and see how the average top-k loss performs on them. One could also try to use other aggregation function over the individual losses to be optimized on a training set (e.g., median, quantiles, OVA, different types of integral? Could you comment on that? Minor comments: - Is the name "ensemble loss" often used? For me, it sounds somehow confusing. - Let \hat{k}^* be tuned on a validation set of size \hat{n}. If we use the entire training set for learinng a final model, should not k^* be appropriately adjusted to reflect the ratio \hat{k}^*/\hat{n}? - line 5 and 307: can combines => can combine After rebuttal: I thank the authors for their response.

Reviewer 2



This paper investigates a new learning setting: optimizing the average k largest (top-k) individual functions for supervised learning. This setting is different from the standard Empirical Risk minimization (ERM), which optimize the average loss function over datasets. The proposed setting is also different from maximum loss (Shalev-Shwartz and Wexler 2016), which optimize the maximum loss. This paper tries to optimize the average top-k loss functions. This can be viewed as a natural generalization of the ERM and the maximum loss. The authors summary it as a convex optimization problem, which can be solved with conventional gradient-based method. The authors give some learning theory analyses of setting on the classification calibration of the Top-k loss and the error bounds of ATk-SVM. Finally, the authors present some experiments to verify the effectiveness of the proposed algorithm. This work is generally well-written with some advantages as follows: 1) The authors introduce a new direction for supervised learning, which is a natural generalization of ERM and the work of (Shalev-Shwartz and Wexler 2016). 2) Some theoretical analyses are presented for the proposed learning setting. 3) The author present a learning algorithm. Cons: 1) Some statements are not clear, for example, top-k loss, which is similar to top-k ranking; more important, ensembles loss gives some impressions for ensemble learning whereas they are totally different. 2) When we used the average top-k loss? I do not think that the authors make clear explanations. Intuitively, the performance (w.r.t. accuracy) of average top-k loss is less than ERM without noise and outliers, while I guess the average too-k loss algorithm may have good performance when deal with noise data and outliers. 3) How to choose the parameter k? The authors use cross-validation in experiments, while there is no some analyses. 4) The authors should present some t-test on the performance on benchmark datasets. I doubt some experimental results. For example, the accuracy for German is about 0.79 for standard learning algorithms. 5) How about the efficiency in comparison with ERM and the work of (Shalev-Shwartz and Wexler 2016).

Reviewer 3



This paper proposed a new ensemble loss (average top-k loss) for supervised learning problems. The average over the k largest individual losses over a training set is used as the objective function for supervised training purpose. The author proposed a convex formulation to implement this idea and formulate the overall problem as a convex optimization which can be solved using gradient methods. The author provides also analysis on how the free parameter k relates to the classification problems and its optimal setting. Similar to standard average loss, the author provide sample complexity bound for ATk based SVM formulation. * on experiments: - it would be better to provide some details on the datasets and why these datasets are selected - if 10 random train/test splits are used, it is better to have std in table 1. - It would be great to include some more comments on why the K-plots in figure 3 have different trends for the four datasets. Is this connected to some property of the dataset? It is not clear to conclude from these plots how to select k in general. - for the regression problem RMSE is used a metric but the author considered both square loss and abs loss. It would be good to use both RMSE and MAE to measure the performance. * Figure 1 has four different synthetic datasets but it is really difficult to parse the information without detailed explanation. It would be more helpful to illustrate the key idea in the introduction by explaining what is the key difference of the four synthetic examples and comments on in which cases the ATk loss makes more sense and helped reduce certain errors.